# EXPECTIGRAD: FAST STOCHASTIC OPTIMIZATION WITH ROBUST CONVERGENCE PROPERTIES

## ABSTRACT

Many popular adaptive gradient methods such as Adam and RMSProp rely on an exponential moving average (EMA) to normalize their stepsizes. While the EMA makes these methods highly responsive to new gradient information, recent research has shown that it also causes divergence on at least one convex optimization problem. We propose a novel method called Expectigrad, which adjusts stepsizes according to a per-component unweighted mean of all historical gradients and computes a bias-corrected momentum term jointly between the numerator and denominator. We prove that Expectigrad cannot diverge on every instance of the optimization problem known to cause Adam to diverge. We also establish a regret bound in the general stochastic nonconvex setting that suggests Expectigrad is less susceptible to gradient variance than existing methods are. Testing Expectigrad on several high-dimensional machine learning tasks, we find it often performs favorably to state-of-the-art methods with little hyperparameter tuning.

## 1 INTRODUCTION

Efficiently training deep neural networks has proven crucial for achieving state-of-the-art results in machine learning (e.g. Krizhevsky et al., 2012; Graves et al., 2013; Karpathy et al., 2014; Mnih et al., 2015; Silver et al., 2016; Vaswani et al., 2017; Radford et al., 2019; Schrittwieser et al., 2019; Vinyals et al., 2019). At the core of these successes lies the backpropagation algorithm (Rumelhart et al., 1986), which provides a general procedure for computing the gradient of a loss measure with respect to the parameters of an arbitrary network. Because exact gradient computation over an entire dataset is expensive, training is often conducted using randomly sampled minibatches of data instead.[1] Consequently, training can be modeled as a *stochastic* optimization problem where the loss is minimized in expectation. A natural algorithmic choice for this type of optimization problem is Stochastic Gradient Descent (SGD) (Robbins & Monro, 1951) due to its relatively cheap computational cost and its reliable convergence when the learning rate is appropriately annealed.

A major drawback of SGD is that its convergence rate is highly dependent on the condition number of the loss function (Boyd & Vandenberghe, 2004). Ill-conditioned loss functions are nearly inevitable in deep learning due to the high-dimensional nature of the models; pathological features such as plateaus, sharp nonlinearities, and saddle points become increasingly probable as the number of model parameters grows—all of which can interfere with learning (Pascanu et al., 2013; Dauphin et al., 2014; Goodfellow et al., 2016; Goh, 2017). Enhancements to SGD such as momentum (Polyak, 1964) and Nesterov momentum (Nesterov, 1983; Sutskever et al., 2013) can help, but they still largely suffer from the same major shortcoming: namely, any particular choice of hyperparameters typically does not generalize well to a variety of different network topologies, and therefore costly hyperparameter searches must be conducted.

This problem has motivated significant research into *adaptive* methods for deep learning, which dynamically modify learning rates on a per-component basis with the goal of accelerating learning without tuning hyperparameters. AdaGrad (Duchi et al., 2011) was an early success in this area that (in its simplest form) divides each step by a running sum of gradient magnitudes, but this can cause its empirical performance to degrade noticeably in the presence of dense gradients. Later methods such as ADADELTA (Zeiler, 2012), RMSProp (Tieleman & Hinton, 2012), and Adam (Kingma & Ba, 2014) remedied this by instead normalizing stepsizes by an exponential moving average (EMA).

---

[1]Training on small minibatches can also improve generalization (Wilson & Martinez, 2003).

Such methods are able to increase their learning rates after encountering regions of small gradients and have enjoyed widespread adoption due to their consistent empirical performance. Unfortunately, the EMA has recently been shown to cause divergence on a certain convex optimization problem (Reddi et al., 2019) that we refer to as the Reddi Problem. This finding has severe implications because it points to an underlying flaw shared by the most widely used adaptive methods.

Recent attempts to resolve this EMA-divergence issue have been unsatisfying. Proposed methods invariably begin with Adam, and then apply a minor adjustment aimed at suppressing divergence. Specifically, they either suddenly or gradually transition from Adam to SGD during training (Keskar & Socher, 2017; Luo et al., 2019), or clip certain quantities in the Adam update rule to limit the EMA's sensitivity (Zaheer et al., 2018; Reddi et al., 2019). While these methods technically do prevent divergence on the Reddi Problem, they fail to address the fundamental issue that the EMA can be unreliable, and they do not advance our theoretical understanding of alternatives that are inherently robust to divergence. Furthermore, by intentionally reducing the responsiveness of Adam, these modifications do not always translate into better empirical performance for problems of practical interest—yet they come at the expense of increased complexity.

The principal objective of this work is to therefore develop a novel adaptive method that provides stronger convergence guarantees than EMA-based methods while retaining fast learning speed. Towards this, we propose the Expectigrad algorithm, which introduces two major innovations: (1) normalization by the arithmetic mean instead of the EMA, and (2) "outer momentum" in which bias-corrected momentum is applied jointly to the numerator and denominator. Expectigrad provably converges on all instances of the Reddi Problem that causes Adam to diverge, and minimizes the function significantly faster than related methods using the same hyperparameters. We also derive a regret bound for Expectigrad that establishes a convergence rate comparable to the best known rate for Adam. Our bounds also indicate that Expectigrad is less susceptible to noisy gradients. Finally, we test Expectigrad by training various neural network architectures with millions of learnable parameters; we show that it consistently outperforms Adam and is competitive with other state-of-the-art methods.

## 2  PRELIMINARIES

We begin with some notational remarks. We always typeset vectors $\mathbf{x}, \mathbf{y}$ in boldface to avoid confusion with scalars $x, y$. We represent the $j$-th component of vector $\mathbf{x}_i$ as $\mathbf{x}_{i,j}$. The Euclidean norm is denoted by $\|\mathbf{x}\|$ and the inner product is denoted by $\langle \mathbf{x}, \mathbf{y} \rangle$. All arithmetic operations can be assumed to be element-wise: e.g. $\mathbf{x} \pm \mathbf{y}$ for addition and subtraction, $\mathbf{xy}$ for multiplication, $\mathbf{x}/\mathbf{y}$ for division, $\mathbf{x}^a$ for exponentiation by a scalar, $\sqrt{\mathbf{x}}$ for square root, and so on.

We now consider the optimization setting that is the focus of this paper. Let $l \colon \mathbb{R}^d \times \mathbb{R}^m \mapsto \mathbb{R}$ be a (nonconvex) function that we seek to (locally) minimize in expectation over some distribution $\mathbb{P}$ on $\Xi \subset \mathbb{R}^m$. Precisely, we must locate a point $\mathbf{x}^* \in \mathbb{R}^d$ with the property $\nabla f(\mathbf{x}^*) = 0$, where $f(\mathbf{x}) := \mathbb{E}[l(\mathbf{x}, \boldsymbol{\xi})]$. All expectations in our work are implicitly taken over $\boldsymbol{\xi} \sim \mathbb{P}(\Xi)$. Direct computation of $\nabla f(\mathbf{x})$ is assumed to be infeasible, but repeated calculation of $\nabla l(\mathbf{x}, \boldsymbol{\xi})$ is permitted. We also assume that $l$ has the following properties:

**Assumption 1.** *$l$ is bounded below.*

**Assumption 2.** *$l$ is Lipschitz continuous: $|l(\mathbf{x}, \boldsymbol{\xi}) - l(\mathbf{y}, \boldsymbol{\xi})| \leq L\|\mathbf{x} - \mathbf{y}\|, \quad \forall \mathbf{x}, \mathbf{y} \in \mathbb{R}^d, \quad \forall \boldsymbol{\xi} \in \Xi$.*

**Assumption 3.** *$l$ is Lipschitz smooth: $\|\nabla l(\mathbf{x}, \boldsymbol{\xi}) - \nabla l(\mathbf{y}, \boldsymbol{\xi})\| \leq L\|\mathbf{x} - \mathbf{y}\|, \quad \forall \mathbf{x}, \mathbf{y} \in \mathbb{R}^d, \quad \forall \boldsymbol{\xi} \in \Xi$.*

Assumption 1 guarantees the existence of a well-defined set of minima (Nocedal & Wright, 2006). While Assumptions 2 and 3 need not share the same Lipschitz constant $L > 0$ in general, we assume that $L$ is sufficiently large to satisfy both criteria simultaneously. These conditions are often met in practice, making our assumptions amenable to the deep learning setting that is the focus of this paper.[2] In Appendix B, we prove three lemmas from Assumptions 2 and 3 that are necessary for our later theorems. With these results, we are ready to present Expectigrad in the following section.

## 3  ALGORITHM

Our motivation for Expectigrad comes from the observation that successful adaptive methods are normalized by an EMA of past gradient magnitudes. This normalization process serves two important

---

[2]We note that the commonly used ReLU activation is unfortunately not Lipschitz smooth, but smooth approximations such as the softplus function (Dugas et al., 2001) can be substituted. This limitation is not specific to our work but affects convergence results for all first-order methods, including SGD.

---

**Algorithm 1** Expectigrad

    Select learning rate $\alpha > 0$                                  $\triangleright$ Default: $\alpha = 10^{-3}$

    Select momentum constant $\beta \in [0, 1)$                         $\triangleright$ Default: $\beta = 0.9$

    Select denominator constant $\epsilon > 0$                        $\triangleright$ Default: $\epsilon = 10^{-8}$

    Initialize parameters $\mathbf{x}_0 \in \mathbb{R}^d$ arbitrarily

    Initialize running sum $\mathbf{s}_0 \leftarrow 0 \cdot \mathbf{x}_0$

    Initialize running counter $\mathbf{n}_0 \leftarrow 0 \cdot \mathbf{x}_0$

    Initialize momentum $\mathbf{m}_0 \leftarrow 0 \cdot \mathbf{x}_0$

    **for** $t = 1, 2, \ldots, T$ **do**

        Compute gradient estimate: $\mathbf{g}_t \leftarrow \nabla l(\mathbf{x}_{t-1}, \boldsymbol{\xi}_t)$ with $\boldsymbol{\xi}_t \sim \mathbb{P}(\Xi)$

        Update running sum: $\mathbf{s}_t \leftarrow \mathbf{s}_{t-1} + \mathbf{g}_t^2$

        Update running counter: $\mathbf{n}_t \leftarrow \mathbf{n}_{t-1} + \mathrm{sign}(\mathbf{g}_t^2)$

        Update momentum: $\mathbf{m}_t \leftarrow \beta \mathbf{m}_{t-1} + (1 - \beta) \frac{\mathbf{g}_t}{\epsilon + \sqrt{\mathbf{s}_t / \mathbf{n}_t}}$         $\triangleright$ Define $\frac{0}{0} = 0$

        Update parameters: $\mathbf{x}_t \leftarrow \mathbf{x}_{t-1} - \frac{\alpha}{1 - \beta^t} \mathbf{m}_t$

    **end for**

    **return** $\mathbf{x}_T$

---

benefits. First, stepsizes become scale invariant; the numerator and denominator are equally affected when the objective function is multiplied by a constant, meaning a particular choice of learning rate is more likely to work well for a wide array of tasks. Second, dividing by the recent gradient magnitude acts as a preconditioner, making the loss function more amenable to first-order optimization. One possible explanation for this is that dividing by the EMA—insofar as it estimates the (diagonal) Fisher information matrix for a log-likelihood function—acts as an approximation to natural gradient descent (Amari, 1997; Pascanu & Bengio, 2013; Kingma & Ba, 2014). A more recent hypothesis is that the EMA equalizes the stochastic gradient noise (Balles & Hennig, 2017), and that this isotropic noise is beneficial for escaping saddle points (Staib et al., 2019).

These properties help explain why EMA-based adaptive methods like RMSProp and Adam are successful on a variety of machine learning problems with relatively little hyperparameter tuning. Nevertheless, the EMA itself has been shown to cause divergence on the Reddi Problem—a synthetic online optimization problem proposed by Reddi et al. (2019). The rapidly decaying nature of the EMA reduces the influence of large yet infrequent gradients, which can cause the optimization to progress in an arbitrarily poor direction. Various related works have proposed to clip terms in the Adam update rule to induce convergence on the Reddi Problem; we discuss them in Section 4.1.

We argue that the approach of constraining EMA-based methods like Adam in an effort to force convergence is counterproductive. Doing so not only risks impacting performance, but it also points to a lack of theoretical understanding. While limiting the adaptation of the EMA may prevent divergence on the Reddi Problem, it does not address the root cause of divergence itself. We are instead interested in algorithms that are automatically robust to rare, high-magnitude gradients that are characteristic of the Reddi Problem. We begin our search by considering the following update rule in which stepsizes are normalized by an arithmetic mean of all gradient samples up to the current timestep $t$:

$$\mathbf{x}_t \leftarrow \mathbf{x}_{t-1} - \alpha \cdot \frac{\mathbf{g}_t}{\epsilon + \sqrt{\frac{1}{t} \sum_{k=1}^{t} \mathbf{g}_k^2}} \tag{1}$$

Here, $\alpha > 0$ is the learning rate and $\epsilon > 0$ is a positive constant that prevents division by zero. Crucially, this update retains the scale-invariance property of EMA-based methods discussed at the beginning of this section: multiplying the gradient by a scalar does not affect the stepsizes (assuming $\epsilon$ is small). Furthermore, the denominator becomes less responsive to new gradients as training proceeds, but never becomes completely static. This is precisely the behavior we want for a method that balances empirical speed with theoretical convergence.

To improve the performance of the algorithm, we will want to make some modifications to (1). First, note that the explicit summation is unnecessary. If we store the sum $\mathbf{s}_t$ at each time $t$, then we can

efficiently compute it from the previous sum: $\mathbf{s}_t \leftarrow \mathbf{s}_{t-1} + g_t^2$. Second, observe that dividing by $t$ to compute the mean could be problematic when gradients are very sparse—commonly the case when using Rectified Linear Unit (ReLU) activations (Nair & Hinton, 2010). Consider what happens when a component of the gradient is zero for a very long time: as $t$ increments unconditionally, the ratio $\frac{s_t}{t}$ approaches zero. When a nonzero component is suddenly encountered, the algorithm will take a large—and potentially catastrophic—step along that dimension. We can mitigate this by introducing an auxiliary variable $\mathbf{n}_t$ that counts the number of nonzero gradients that have occurred. Note that we can concisely update $\mathbf{n}_{t-1}$ by simply adding the element-wise sign of the squared gradient: $\mathbf{n}_t \leftarrow \mathbf{n}_{t-1} + \text{sign}(\mathbf{g}_t^2)$. Putting these two changes together (and defining $\frac{0}{0} = 0$),

$$\mathbf{x}_t \leftarrow \mathbf{x}_{t-1} - \alpha \cdot \frac{\mathbf{g}_t}{\epsilon + \sqrt{\frac{\mathbf{s}_t}{\mathbf{n}_t}}} \tag{2}$$

This new algorithm tracks a per-component arithmetic mean where each component may have a distinct number of counts, thereby ignoring sparsity in the optimization problem. Thus, with some additional memory, we have made the update highly efficient and less prone to numerically instability.

Finally, we can also apply momentum to help accelerate learning when the gradient signal is weak. Traditionally, momentum is first applied to the gradient estimator, and then is scaled by the adaptive method (e.g. Adam). In our work, we propose to reverse the order by computing the adaptive steps first and then applying momentum to them. This is important to satisfy the superposition principle: the momentum step "scheduled" at the current timestep is not re-scaled by future changes in the denominator. Letting $\beta \in [0, 1)$ be the momentum constant, we arrive at our final update:

$$\mathbf{x}_t \leftarrow \mathbf{x}_{t-1} - \frac{\alpha}{1 - \beta^t}\mathbf{m}_t \quad \text{where} \quad \mathbf{m}_t = \beta\mathbf{m}_{t-1} + (1 - \beta)\frac{\mathbf{g}_t}{\epsilon + \sqrt{\frac{\mathbf{s}_t}{\mathbf{n}_t}}} \tag{3}$$

We call this "outer" momentum as it applies to the entire update rule and not just the numerator. The coefficient $1/(1-\beta^t)$ is a bias-corrective factor introduced by Kingma & Ba (2014). We refer to (3) as Expectigrad and formally present it in Algorithm 1. While we have only intuitively justified our design choices so far, we will spend the next few sections rigorously analyzing their theoretical and empirical implications.

## 4 COMPARISON WITH RELATED METHODS

Expectigrad eschews the familiar EMA in favor of the arithmetic average, and utilizes a newly formulated momentum that incorporates adaptive stepsizes. It is not immediately apparent that these particular choices are better; this section is devoted to illuminating the benefits of these substitutions by comparing them with other approaches in the literature.

### 4.1 ARITHMETIC MEAN NORMALIZATION

Unlike existing adaptive methods, Expectigrad is normalized by a non-exponential mean. Perhaps the most salient concern with this is that the denominator becomes insensitive to new gradients as the algorithm progresses. It turns out that this behavior is beneficial, not harmful. To see this, we formally introduce the Reddi Problem, the online optimization problem from Reddi et al. (2019):

$$h_t(x) = \begin{cases} Cx & \text{if } t \equiv 0 \pmod{N} \\ -x & \text{otherwise} \end{cases} \tag{4}$$

It is assumed that $C > N - 1$ and therefore moving towards $-\infty$ minimizes this function.[3] In-triguingly, Adam incorrectly proceeds towards $+\infty$ when $N = 3$ when its EMA adapts too quickly (Reddi et al., 2019). This explains why its EMA constant $\beta_2$ is typically very close to 1 in practice. In Figure 1, we replicate the experiment from Reddi et al. (2019) and show that Expectigrad can indeed minimize $h_t(x)$ while Adam cannot. RMSProp and ADADELTA diverge too because they utilize EMAs for normalization.

We also compare against recent methods AMSGrad (Reddi et al., 2019) and Yogi (Zaheer et al., 2018) that have been proposed to fix divergence by modifying the Adam update rule. Specifically, AMSGrad

---

[3]The domain of $h_t$ was restricted to $[-1, 1]$ in Reddi et al. (2019), making the optimal solution $x^* = -1$. We will not impose this restriction; it therefore suffices to show that $\lim_{t \to \infty} x_t = -\infty$ to prove non-divergence.

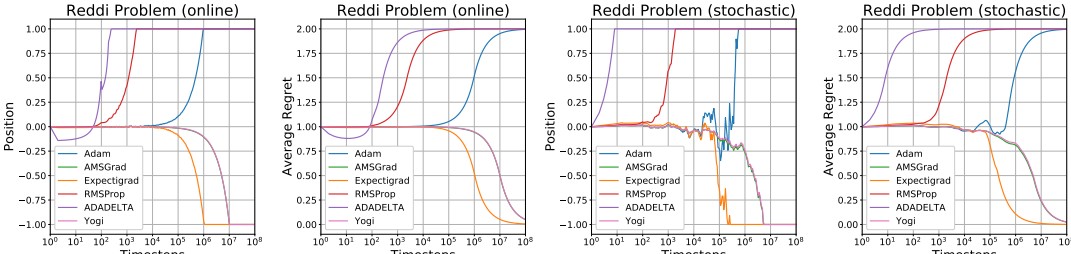

Figure 1: We replicated the synthetic convex optimization problem from Reddi et al. (2019) that causes Adam to diverge. We also tested RMSProp and ADADELTA, two other EMA-based methods, and found that they diverge too. Compared to AMSGrad and Yogi, Expectigrad minimizes both functions about 10 times faster despite using the same hyperparameters. (Note that AMSGrad is partially occluded by Yogi in the graphs.) We clipped the domain to $[-1, 1]$ for graphical purposes.

normalizes stepsizes using the maximum past EMA value, thereby enforcing a monotonically non-decreasing denominator. Yogi similarly limits the effective rate of change of its EMA. While both of these approaches succeed in preventing divergence on the Reddi Problem, simply constraining the EMA in this manner is not an ideal solution; it sacrifices not only scale invariance but also the ability to respond quickly to sudden topological changes when necessary. Figure 1 illustrates the major consequence of these harmful effects: slow learning. Despite having identical hyperparameters, AMSGrad and Yogi require nearly 10 times more experience than Expectigrad does to reach $x = -1$.

In addition to our experiments, we can prove analytically that Expectigrad will never diverge on the Reddi Problem. For simplicity, we let $\beta = 0$ for all theorems in our paper.

**Theorem 1.** *(Non-divergence of Expectigrad on the Reddi Problem.) Let $h_t$ be the function defined in (4) and let $\{x_t\}_{t=1}^{\infty}$ be a sequence of points generated by Expectigrad. For any integer $N \geq 2$ and any real number $C > N - 1$, $\lim_{t \to \infty} x_t = -\infty$. Hence, Expectigrad never diverges on this problem.*

We relegate the proof to Appendix C. In brief, we show that the displacement of Expectigrad over any period of $h_t(x)$ is bounded above by a negative constant. This implies that the total displacement of Expectigrad after an arbitrarily large number of periods must approach $-\infty$. An important corollary of Theorem 1 is that there exists at least one problem where Adam diverges but Expectigrad does not.

That Expectigrad always converges on this problem holds practical significance. The Reddi Problem mimics a dataset where a few training samples are highly informative but rare—a setting that commonly arises in practical machine learning problems. While EMA-based methods tend to diminish the importance of these rare samples, and clipped-EMA methods tend to learn slowly, Expectigrad converges both quickly and reliably.

### 4.2 OUTER MOMENTUM

We now study the effect of outer momentum on learning speed by conducting experiments on the classic MNIST handwriting recognition task (LeCun et al., 1998). The training and test sets respectively consist of 60,000 and 10,000 grayscale images of size $28 \times 28$ that must be classified according to which digit between 0 and 9 they contain. Following Kingma & Ba (2014), we trained a 2-layer fully connected network with 1,000 ReLU units per layer.

In our first experiment, we analyze the impact of substituting various forms of momentum into Expectigrad: classic "inner" momentum (Polyak, 1964), bias-corrected "inner" momentum (Kingma & Ba, 2014), and our bias-corrected "outer" momentum in (3). Appendix A provides their update equations for reference. For each variant, we tested a range of $\beta$-values and measured the training loss after exactly one epoch of training (Figure 2, left). This offers insight into how different momenta impact performance early in training. We see that bias correction is crucial for fast learning, evidenced by the poor performance of the uncorrected baseline. Simultaneously, for every choice of $\beta$, our bias-corrected outer momentum performs at least as well as bias-corrected inner momentum. Notably, the performance gap widens quickly as $\beta$ approaches 1. As discussed earlier, we attribute the improvement of outer momentum to its preservation of linear superposition. Formally, linear superposition holds when $\sum_{t=1}^{\infty} \boldsymbol{\delta}_t = \sum_{t=1}^{\infty} \mathbf{m}_t$ for momentum $\mathbf{m}_t = \beta \mathbf{m}_{t-1} + (1-\beta)\boldsymbol{\delta}_t$. Outer

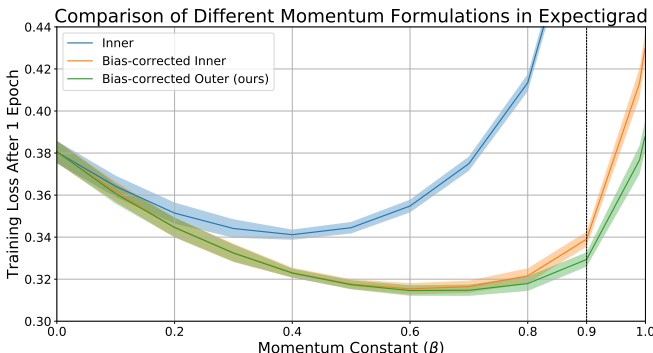
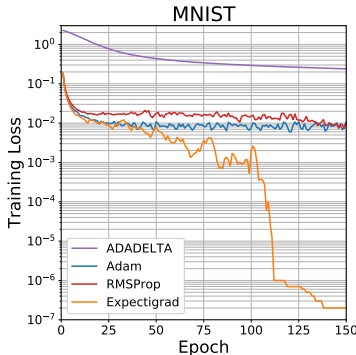

Figure 2: Using the MNIST dataset, we explored the effects of different momenta on Expectigrad by training for a single epoch (left). We swept the momentum constant $\beta \in [0, 1)$ for three variants: inner (Polyak, 1964), bias-corrected inner (Kingma & Ba, 2014), and bias-corrected outer (ours). The dashed vertical line represents $\beta = 0.9$, the commonly used default value for Adam and its extensions. Outer momentum always outperforms inner momentum, and the improvement becomes highly pronounced as $\beta \to 1$. Results were averaged over 10 runs.

momentum clearly obeys this because all adaptation occurs before the momentum update; on the other hand, inner momentum adapts $\mathbf{m}_t$ directly, implicitly re-scaling previous impulses and therefore violating the equality.

Because the first epoch of training may not be indicative of overall performance, we next test Expectigrad's performance over 150 epochs of training. We compared against three EMA-based methods—RMSProp, ADADELTA, and Adam—using their default hyperparameter values from TensorFlow (Abadi et al., 2016) (see Appendix A). For Expectigrad, we adopted Adam's default values: $\alpha = 10^{-3}$, $\beta = 0.9$, $\epsilon = 10^{-8}$. Expectigrad achieves a training loss several orders of magnitude below that of the other methods (Figure 2, right). These results demonstrate that Expectigrad performs very well empirically despite borrowing hyperparameter values from an existing method.

## 5  CONVERGENCE ANALYSIS

Theorem 1 asserts that Expectigrad cannot diverge on the Reddi Problem, but it does not guarantee that Expectigrad converges for a wide class of functions. Our next step is to establish regret bounds for the general case of a smooth, nonconvex function. We define the cumulative regret at time $T$ in terms of the gradient norm: $R(T) = \sum_{t=1}^{T} \mathbb{E}\left[\|\nabla f(\mathbf{x}_t)\|^2\right]$. Under analysis similar to that of Zaheer et al. (2018), we arrive at the next theorem.

**Theorem 2.** *(Regret Bound for Expectigrad.) Let $l$ be a function satisfying Assumptions 1, 2, 3 and let $\{\mathbf{x}_t\}_{t=1}^{T}$ be a sequence of points in $\mathbb{R}^d$ generated by Expectigrad with learning rate $\alpha \leq \frac{\epsilon}{L}$ and denominator constant $\epsilon > 0$. Given gradient variance $\sigma^2$ and minibatch size $b \geq 1$, the average regret incurred by Expectigrad has the following upper bound:*

$$\frac{1}{T}R(T) < (\epsilon + L)\left[\frac{f(\mathbf{x}_0) - f(\mathbf{x}^*)}{\alpha T} + \frac{2L}{\epsilon^2 \sqrt{T}}\left(L^2 + \frac{\sigma^2}{b}\right) + \frac{\sigma^2}{2\epsilon b}\right] = O\left(\frac{1}{T} + \frac{1}{\sqrt{T}} + \frac{1}{b}\right)$$

The proof is in Appendix D. Like other stochastic first-order methods, Expectigrad approaches but does not truly reach a stationary point due to the variance of the gradient estimator. The regret vanishes when utilizing a time-increasing batch size (Zaheer et al., 2018) or annealed learning rate.

Comparing this bound to those of Adam and Yogi derived in Zaheer et al. (2018) offers some insights. First, Expectigrad achieves the same asymptotic convergence rate as these methods do. Second, Expectigrad converges for a much larger set of hyperparameter values; whereas Adam and Yogi both require $\alpha \leq \frac{\epsilon}{2L}$ and $\epsilon \geq 4L\sqrt{1 - \beta_2}$, Expectigrad needs only $\alpha \leq \frac{\epsilon}{L}$ and $\epsilon > 0$. Thus, Expectigrad entirely removes the unrealistic limitation on $\epsilon$ and doubles the maximum permissible value of $\alpha$.

Third, the right-most term of our bound in Theorem 2 is $O(\frac{1}{\epsilon})$, while the analogous term for Adam and Yogi is $O(\frac{1}{\epsilon^2})$. This term accounts for the irreducible regret due to the statistical variance of the gradient estimate. The denominator constant $\epsilon$ is typically chosen to be very small in practice, which

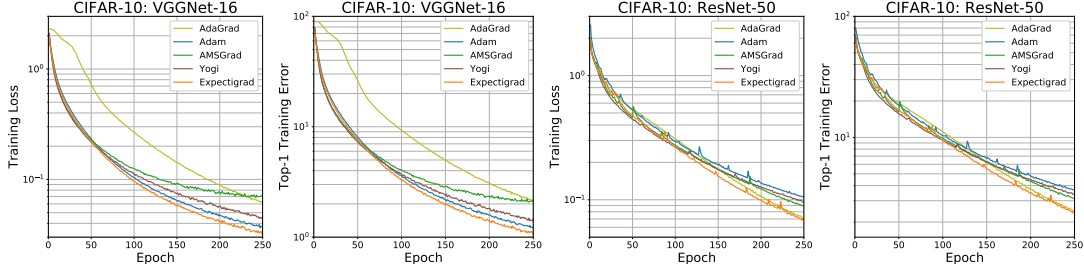

Figure 3: We compared five adaptive methods on CIFAR-10 by training two large convolutional models, VGGNet-16 (left, center-left) and ResNet-50 (center-right, right). Hyperparameters were chosen by a small grid search (Appendix A). Expectigrad outperformed the baseline methods for both models with respect to training loss and top-1 error rate. Results were averaged over 10 runs.

implies $\frac{1}{\epsilon} \ll \frac{1}{\epsilon^2}$ and in turn suggests that Expectigrad is less susceptible to gradient noise than Adam and Yogi are. These results may hold empirical significance, which we evaluate in Section 6.

## 6 EXPERIMENTS

In this section, we conduct a variety of experiments to study the empirical performance of Expectigrad. Details for preprocessing, augmentation, and hyperparameter selection are provided in Appendix A.

**CIFAR-10** We begin with a challenging image classification benchmark, the CIFAR-10 dataset (Krizhevsky et al., 2009), where colored $32 \times 32$ images of various objects must be classified into one of 10 categories. The training and test sets have 50,000 and 10,000 images, respectively. We trained two models for 250 epochs: a 16-layer VGGNet (Simonyan & Zisserman, 2014) with 33.6 million parameters and a 50-layer ResNet (He et al., 2016) with 23.5 million parameters. We conducted a small grid search for $\alpha$- and $\epsilon$-values. In Figure 3, we compare five adaptive methods: AdaGrad, Adam, AMSGrad, Yogi, and Expectigrad. For both network architectures, Expectigrad achieves lower training loss and error than the other methods do, demonstrating that Expectigrad is competitive with the best methods from the deep learning literature.

**English-Vietnamese Translation** The IWSLT'15 English-Vietnamese translation task features a training set of over 133,000 sentence pairs and a test set of 1,268 sentence pairs. We implemented a 4-layer, 1024-unit LSTM (Hochreiter & Schmidhuber, 1997) and a Transformer (Vaswani et al., 2017) using the `tensor2tensor` library Vaswani et al. (2018). These models are significantly different from the feedforward architectures we have tested so far, and are both highly non-trivial to optimize. We compared Expectigrad against four "Adam-like" methods: Yogi, AMSGrad, Adafactor (Shazeer & Stern, 2018), and Adam. The final BLEU scores (Papineni et al., 2002) on the test data are reported in Table 1. Expectigrad, Yogi, and AMSGrad all performed well on both tasks. Expectigrad did the best with the Transformer, and achieves scores near the state-of-the-art results from Zaheer et al. (2018) without any learning rate annealing. Despite training for twice as long with the LSTM, these three methods achieved only about half of the Transformer's score— likely because the LSTM model is less expressive. Adafactor performed somewhat worse for both models but still achieved decent performance. To our surprise, Adam failed completely at both tasks. We speculate that its poor performance could be related to its divergent behavior on the Reddi Problem; perhaps these translation tasks also exhibit large, rare gradients that are critical for learning. This would explain why Expectigrad, Yogi, and AMSGrad all outperform Adam so significantly, but further empirical evidence would be needed to conclusively determine this.

**ImageNet** For our final experiment, we compare Expectigrad and Yogi on the massive ILSVRC 2012 classification task. This dataset features more than 1.3M training images and 50,000 validation images from the ImageNet database (Russakovsky et al., 2015) divided into 1,000 classes. We trained a MobileNetV2 (Sandler et al., 2018) for 300 epochs. We plot training loss, top-1 training error rate, and top-5 validation error rate in Figure 4. While both optimization methods perform reasonably well, Expectigrad achieves lower error rates on the training and test data than Yogi does. This is in spite of using the recommended hyperparameters for Yogi. We note that a specific learning rate annealing schedule for Yogi was used by Zaheer et al. (2018) to outperform a highly-tuned RMSProp implementation; our results here suggest that Yogi's empirical success in that instance may be due

|  | LSTM | | Transformer | |
| --- | --- | --- | --- | --- |
| Method | Cased | Uncased | Cased | Uncased |
| Expectigrad | **13.68 ± 0.28** | **14.18 ± 0.27** | **28.33 ± 0.17** | **29.12 ± 0.16** |
| Yogi | **13.86 ± 0.20** | **14.39 ± 0.22** | **28.30 ± 0.16** | **29.07 ± 0.15** |
| AMSGrad | **14.00 ± 0.23** | **14.50 ± 0.25** | **28.27 ± 0.31** | **29.04 ± 0.29** |
| Adafactor | 12.65 ± 0.15 | 13.11 ± 0.13 | 24.36 ± 0.53 | 25.05 ± 0.52 |
| Adam | 0.03 ± 0.01 | 0.06 ± 0.01 | 0.30 ± 0.09 | 0.38 ± 0.09 |

Table 1: BLEU scores for the IWSLT'15 English-Vietnamese translation task with LSTM and Transformer. Results were averaged over 10 random seeds with standard deviation reported. Bold values represent the best model scores that overlap within one standard deviation. While Adam could not succeed at either task, Expectigrad achieved performances competitive with AMSGrad and Yogi.

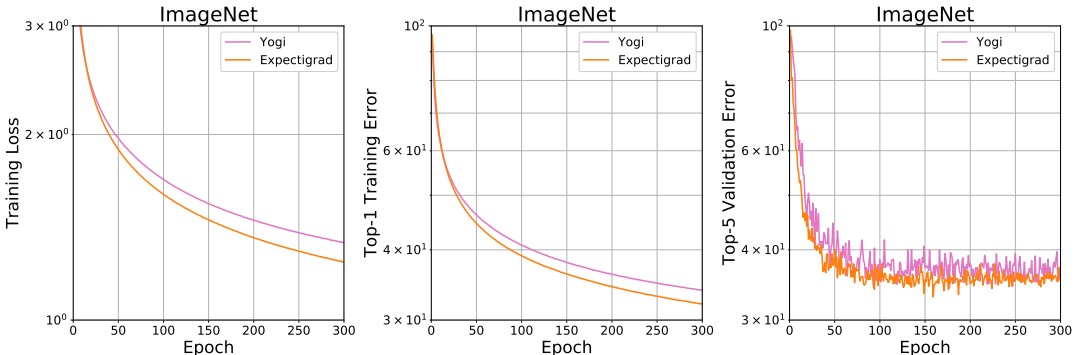

Figure 4: We compared Expectigrad and Yogi training a MobileNetV2 on the ImageNet classification task. Despite using the $10\times$ larger learning rate for Yogi recommended by Zaheer et al. (2018), Expectigrad achieves a lower loss and training/validation error rates in the same amount of training. Results were averaged over 3 runs.

more to its initially large learning rate and a sophisticated annealing schedule rather than its particular adaptation scheme.

**Summary** We tested Expectigrad on a diverse array of deep learning architectures: fully connected (Section 4); convolutional (with and without residual connections); recurrent; and attention networks. Despite significant differences in these network topologies, Expectigrad was among the top-performing methods for each task—especially for the CIFAR-10 experiments, where Expectigrad was the best method with respect to both training loss and top-1 error. In the English-Vietnamese translation tasks, Adam was unsuccessful in training either language model while Expectigrad achieved results competitive with the most recent methods from the literature. Finally, the large-scale ImageNet experiment affirms that Expectigrad's arithmetic mean remains effective even when training for millions of steps.

## 7 CONCLUSION

We introduced Expectigrad, an adaptive gradient method that normalizes stepsizes with an arithmetic mean instead of an EMA and computes momentum jointly between its numerator and denominator. We proved that Expectigrad converges for smooth (possibly nonconvex) functions including the Reddi Problem where EMA-based methods like Adam, RMSProp, and ADADELTA fail. Expectigrad performed competitively across a wide variety of high-dimensional neural network architectures, often outperforming state-of-the-art methods. Our machine translation experiments revealed practical problems where Adam diverges yet Expectigrad achieves very strong performance, providing potential affirmation of our theoretical predictions. Ultimately, Expectigrad's fast performance refutes any notion that adaptive methods must rely on an EMA to be responsive to recent gradient information.

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
