# OpenReview forum: "Expectigrad: Fast Stochastic Optimization with Robust Convergence Properties"
_ICLR.cc/2021/Conference — Reject_

### Official Review · AnonReviewer2 · 2020-10-26
**The paper proposes a new adaptive method for stochastic optimization with convergence guarantees. Experimental results on two tasks viz., image classification and language translation are provided to show the benefit of the proposed algorithm.**

**Rating:** 5
**Confidence:** 5

**Review:**

Summary: The paper proposes a new adaptive method for stochastic optimization with convergence guarantees. Experimental results on two tasks viz., image classification and language translation are provided to show the benefit of the proposed algorithm.

Strength:

+ The paper proposes an adaptive stochastic optimization scheme called Expectigrad for training deep neural networks. Expectigrad is shown to converge (to a stationary point) at the same rate as other recently introduced adaptive methods such as Adam, and Yogi. To motivate their algorithm, the paper studies the Reddi problem introduced in Reddi et al., and show that there exists an averaging scheme to prevent divergence in that "class" of problems. The paper as such motivates the algorithm in the general case first and then applies the arithmetic mean normalization to the Reddi problem, but I find the other way also reasonable, and intuitive.
+ Apart from the averaging scheme, they also propose a novel momentum term that automatically achieves scale invariance, that is, (if I understand correctly) the step size is invariant to coordinate transformations of the loss function. They provide a detailed theoretical and empirical justification for the momentum term.

Weakness:

- While the novelty of the algorithm proposed in clear, the paper is hard to read in certain parts, and I'm not sure if they are correct. For example, in Page 2, the paper mentions that Zaheer  et al, Reddi et al fail to address the "fundamental" issue, and I'm not sure if the paper specifies the issue explicitly anywhere. To this particular point, the paper misses an important reference arXiv:1910.07454 where they show that exponential learning rates may be possible indeed. Perhaps some discussion on this in the paper would make it clear to the reader.  Second, above equation (3), the authors mention that it is important to satisfy the "superposition" principle without any reference or explanation. Third, in section 4.1 (above Theorem 1), I'm not sure what "topological" changes are? In the function with respect to the parameters or data? Indeed, this is an important distinction that needs to be clarified.

- The experiments showing the benefits of Expectigrad is far from convincing. There is a discrepancy in the results being reported. For example, in CIFAR10 experiments, training loss (and training error) are reported whereas for the language translation, BLEU scores on the test set are reported, and in imagenet experiments, again training (and validation) measures are reported. The experiments were averaged over 10 runs in CIFAR10 experiments whereas 3 runs in Imagenet experiments whereas averaging is not discussed for other simulations, MNIST experiments.  Do the training curves provided have confidence? I mean the results are averaged, so providing confidence interval is easy and provides more information about the algorithm.

After response: Thanks for the clarifications. The proposed method is a modification of existing averaging schemes to schedule learning rate, but more experimental evaluations are required to determine the benefits of algorithm.

---

> ### Author Response · Authors · 2020-11-19
> **Expectigrad demonstrates strong empirical performance compared to Adam, RMSProp, AdaGrad, and others**
>
> Thank you for the review.
>
> We would like to emphasize that our experiments do show convincing benefits of Expectigrad. This is particularly evident on the MNIST and CIFAR-10 datasets, where Expectigrad outperforms several well-known methods including Adam, RMSProp, and AdaGrad. Furthermore, on the challenging Transformer machine translation experiment, Expectigrad achieved performance competitive with some of the latest adaptive methods. On the other hand, Adam completely failed to train the model.
>
> The "discrepancies" in the reported results that you noted are due to the wide array of machine learning benchmarks we tested in our paper. We aimed to report results that are consistent with related works; this meant that the image recognition benchmarks often reported training loss and error, while the machine translation benchmarks reported test BLEU scores. This allowed us to compare our results directly with the literature.
>
> We also thank you for bringing to our attention parts of the paper that may need clarification:
>
> (1) The "``fundamental" issue we discuss on page 2 is "that the EMA can be unreliable," in regard to methods like AMSGrad and Yogi that seek to prevent the divergence of Adam by clipping quantities in its update rule rather than seeking alternatives to the EMA. This is different from the "exponential learning rate schedule" in the arXiv reference you have provided.
>
> (2) The mathematical motivation for preserving the "superposition" principle is elaborated in Section 4.2. We also briefly mention this in earlier in Section 3, which may have led to the confusion.
>
> (3) In Section 4.1, "topological" could likely be replaced with "topographical." We were referring to changes in the local loss landscape encountered by the algorithm as training progresses, meaning that these changes are with respect to the model parameters. (The data distribution is assumed to be stationary.)

---

### Official Review · AnonReviewer1 · 2020-10-28
**Too much simplifications for the analysis take away the novelties in the algorithm, making the contribution questionable**

**Rating:** 3
**Confidence:** 4

**Review:**

This paper proposes Expectigrad, which is a new optimizer for nonconvex optimization. The main idea is to consider arithmetic mean of squared gradients instead of exponential moving average and to use a normalization factor that takes into account the number of nonzeros observed during the run of the algorithm, for each component. The algorithm is analyzed for solving smooth nonconvex optimization and its practical performance is investigated.

In terms of the strengths of the paper, I found the idea of using normalization $n_t$ depending on the sparsity interesting. However, the idea of using $s_t$, which is given in eq. (1) seems to be not novel. In particular, this step is already used in Adagrad based methods. For example, AdamNC method in Reddi et al., 2019 already shows that with such an update for second moment estimate, one can obtain convergence. This algorithm is also analyzed by Chen et al., 2018 in the nonconvex setting, which seems to be missed by the authors.

In terms of the analysis, the authors make the simplifying assumptions of $\beta=0$ in page 5 and ignoring sparsity in page 16, which basically converts the algorithm to Adagrad. Can the authors clarify this connection and let me know if I am missing something? Therefore, I am not sure if the analysis brings new results that are not known in the literature. Such simplifications take away the novel parts of the algorithm and makes it impossible for the authors to show the effect of their idea in theory. Therefore the potential benefit of the method becomes purely experimental. For example, it is known in the literature that the extension of analyses for including nonzero (especially constant) $\beta$ is not always easy and one needs to be careful [1].

Moreover, the author's usage of the term "regret" is not correct. The authors use the expected gradient norm as the optimality measure, which is the standard one for nonconvex optimization. Regret is used for online optimization, which is more general than stochastic optimization which is considered in this paper.

Next, the authors analyze the algorithm in Thm 2 with increasing mini-batch sizes. However, Zaheer et al., 2018 shows that with increasing (or depending on the number of iterations $T$) mini-batch sizes, Adam (which is non-convergent in general) also works. Therefore,  I think this setting is not suitable since increasing mini-batch sizes shadow most of the theoretical challenges. It is better to analyze the standard setting without increasing mini-batch sizes.

In practice, the comparisons with Adagrad and AdamNC are omitted. Moreover, it seems that the improvement of Expectigrad compared to SOTA is not significant.

Therefore, in terms of algorithmic ideas and the novelties in theoretical analysis, this paper does not meet the bar for ICLR. In particular, the simplifying assumptions that the authors make for the algorithm ($\beta=0$ and no sparsity) essentially takes away the interesting parts of the proposed algorithm, and making the theoretical analysis not consistent with the algorithm used in practice. Second, the algorithm needs to use increasing mini-batch sizes (the rate in Thm. 2 has $1/b$ as an additive term), which is another oversimplified assumption for the analysis, many adaptive algorithms are proven to converge without this assumption. Empirical merit of the method is also marginal and some important comparisons are missing. As a result, I am voting for rejection.

Chen et al., 2018: Xiangyi Chen, Sijia Liu, Ruoyu Sun, and Mingyi Hong. On the convergence of a class of adam-type algorithms for non-convex optimization. ICLR, 2019.

[1] A New Regret Analysis for Adam-type Algorithms, Alacaoglu, Malitsky, Mertikopoulos, Cevher, ICML 2020.


======== after discussion phase =======

The main drawbacks of the paper remain after the discussion. Mainly, the analysis is too simplistic which basically ignores the new aspects of the algorithm and its comparison with well-known methods is unclear. Therefore, I keep my score.

---

> ### Author Response · Authors · 2020-11-19
> **Expectigrad outperforms AdaGrad in our experiments**
>
> Thank you for the review.
>
> Although we let $\beta=0$ and ignore sparsity in our proofs, the analysis is not identical to that of AdaGrad. In another comment above, we show why this is the case mathematically (https://openreview.net/forum?id=BvrKnFq_454&noteId=lrsQYnokN2A). In brief, AdaGrad is normalized by a sum of past gradients, causing the learning rates to be monotonically non-increasing. On the other hand, Expectigrad is normalized by the mean of past gradients, so there is no guarantee of this monotonicity. It is therefore not obvious that this algorithm would converge, but our analysis shows that it does.
>
> We would further like to point out that we do compare Expectigrad to AdaGrad in our VGGNet and ResNet experiments on CIFAR-10, where we outperform it by a significant margin. The other experiments on MNIST, ImageNet, and IWSLT'15 also demonstrate promising performance in comparison to state-of-the-art methods.
>
> Finally, we thank you for bringing the AdamNC extension to our attention; we see how it is relevant to Expectigrad and we can certainly discuss it in our paper.

---

### Official Review · AnonReviewer3 · 2020-10-28
**Marginal contribution**

**Rating:** 4
**Confidence:** 3

**Review:**

The paper proposes Expectigrad, which replaces the exponential moving average in Adam and RMSProp with averaging over all historical gradients. It shows theoretically and empirically that it outperforms the state-of-the-art stochastic optimization schemes. I have the following questions for the authors:


1. Could you please elaborate more on why average over all historical data can be better than EMA? It's not fully convincing to me. Imagine the case when I first have very small gradients (but non-zero so n_t also adds up) for a long time, and suddenly I've got a gradient that is much larger. In this case, the gradient step would be arbitrarily large and brings a lot of instability. With this example, I can easily construct some scenarios that lead to divergence.



Why does this not conflict with Theorem 2? It seems that the  Lipschitz gradient assumption helps. But the Adam paper does have a convergence guarantee without the Lipschitz gradient assumption. So there does exist some scenario where Adam converges while Expertigrad diverges.


2. At the end of the introduction, the authors claimed that 'Expectigrad provably converges on all instances of the Reddi Problem that causes Adam to diverge, and minimizes the function significantly faster than related methods using the same hyperparameters.' Does that mean you are comparing different algorithms with the same hyperparameters instead of doing grid-search for each algorithm separately?



3. In the algorithm description and equation (1), it does not make sense to have a square of a vector: what does it mean? Is that element-wise square or two norm of the vector? Is s_t a scalar or vector? It is also weird to have the scalar n_t the same boldface as the other vectors.


I read the author feedback and decide to keep my score.

---

> ### Author Response · Authors · 2020-11-19
> **Our theory and experiments demonstrate that the non-exponential moving average helps with convergence**
>
> Thank you for the review. We will address your questions below:
>
> (1) Our paper demonstrates that using the average over all historical data can be better than the EMA because it reduces the responsiveness of the optimization method over time. In pathological cases, such as the Reddi Problem we discussed in our paper, the EMA's ability to change too quickly can cause the algorithm to move in the wrong direction and eventually diverge. By replacing the EMA with an unweighted average, Expectigrad's adaptation becomes less sensitive over time, eventually behaving more like a preconditioned form of SGD and taking unbiased steps on average.
>
> You are correct that the hypothetical scenario that you propose (many small gradients followed by a big gradient) could produce a large step. However, this effect is not unique to Expectigrad and other algorithms like Adam and RMSProp would behave similarly. Furthermore, a large step does not immediately cause divergence; subsequent steps will be drastically reduced in magnitude, and on average the optimization will make progress towards a stationary point. This is consistent with our theory because of the restrictions on $\alpha$ and $\epsilon$ we impose in Theorem 2, which guarantee that the maximum step taken by the algorithm is no more than $\frac{1}{L}$ and will not cause divergence.
>
> Finally, we note that the Adam paper's convergence proof focuses exclusively on convex functions, and the authors anneal the learning rate over time, which is likely why the Lipschitz gradient assumption is not used.
>
> (2) For the Reddi Problem, we compare several mean-normalized algorithms with the same learning rate. This means that the only differences between these methods is how their denominators are computed, which allows us to explore how their adaptation schemes affect their performance. On the larger datasets, we tune the hyperparameters and select those with the best performance.
>
> (3) In the first paragraph of Section 2, we state that all operations can be assumed to be element-wise; therefore, the square of a vector should be interpreted as the element-wise square. The variables $\mathbf{s}_t$ and $\mathbf{n}_t$ are both boldface because they are vectors, which respectively track the element-wise sum and element-wise density count of the past gradients.

---

### Official Review · AnonReviewer4 · 2020-10-28
**Adam-type step size adjustment on the fly**

**Rating:** 5
**Confidence:** 3

**Review:**

Summary: This paper proposes the Expectigrad algorithm that normalizes the exponential moving average (EMA) of first moments on the fly. This avoids normalizing historical gradients by future gradients. The normalization factor is an unweighted average, instead of an EMA, of the historical second moments. For the special case where the EMA constant of the first moment is zero, the paper shows that Expectigrad converges to the optimum on the online convex problems proposed by Reddi et al. (2018) for which the vanilla ADAM fails to converge. For general stochastic smooth convex problems with bounded stochastic gradients, the paper shows that the convergence rate of mini-batch Expectigrad is $O(1 / \sqrt{T} + 1 / b)$ where $b$ is the mini-batch size.

Pros:
(1) The idea of normalizing on the fly is interesting because it is more sensitive when the gradient dynamic is non-stationary.

(2) The algorithm performs well on the examples considered in the paper.

Concerns:
(1) Theorem 1 assumes $\beta = 0$ "for simplicity". I think this is too weak to justify the algorithm. When $\beta = 0$, the algorithm is almost identical to AdaGrad, except that AdaGrad does not take the sparsity into account and sets $n_{t} = t$. On the Reddi problem, the gradient is not sparse, and thus Expectigrad with $\beta = 0$ is equivalent to AdaGrad, if I am not mistaken. Reddi et al. (2018) has already shown that AdaGrad does not diverge on the counterexamples. In order to justify the algorithm, it is necessary to prove a similar result for the case $\beta > 0$, since it is set to be $0.9$ in the experiments. Reddi et al. (2018) have results of this kind (which only requires $\beta < \sqrt{\beta_2}$ where $\beta_2$ is the EMA constant of the second moment).

(2) The paper claims (in the bottom of page 6) that Expectigrad has strictly better complexity than Yogi. I do not see why this is the case. The convergence rate of Yogi is $O(1 / T + 1 / b)$ (their Corollary 4) while that of Expectigrad if $O(1 / \sqrt{T} + 1 / b)$. The latter is worse. To take one step further, in order to have $E ||\nabla f(x)||\le \epsilon$, we need to set the convergence rate as $\epsilon^2$ since the bound is proved for $E ||\nabla f(x)||^2$. For Yogi, the mini-batch size $b$ needs to be $O(1 / \epsilon^2)$ and the number of iterations $T$ needs to be $O(1 / \epsilon^2)$ as well. So the overall complexity is $O(1 / \epsilon^4)$ which matches the complexity of SGD. However, for Expectigrad, the mini-batch size $b$ needs to be $O(1 / \epsilon^4)$ and the number of iterations $T$ needs to be $O(1 / \epsilon^2)$. The overall complexity is $O(1 / \epsilon^6)$, which is much higher than SGD. I do not understand why Expectigrad is better than Yogi or even SGD.

(3) The footnote in page 2 states that "This limitation is not specific to our work but affects convergence results for all first-order methods, including SGD". This is incorrect. Under assumption 2 of the bounded stochastic gradient, SGD converges for non-smooth functions.

---

> ### Author Response · Authors · 2020-11-19
> **Expectigrad's complexity is not worse than Yogi's**
>
> Thank you for the review. Addressing your three concerns,
>
> (1) We agree that analyzing the convergence of Expectigrad for the case of $\beta > 0$ would be interesting. However, our analysis for $\beta=0$ is still novel, since this case is not identical to AdaGrad. In particular, AdaGrad is normalized by a sum of past gradients, whereas Expectigrad is normalized by the mean of past gradients. We can see that Expectigrad with $\beta=0$ and non-sparse updating takes steps $\alpha \cdot \frac{g_t}{\epsilon + \sqrt{s_t / t}} = \alpha \sqrt{t} \cdot \frac{g}{\epsilon \sqrt{t} + \sqrt{s_t}}$, which differs from AdaGrad because $\alpha$ and $\epsilon$ are both scaled by $\sqrt{t}$. It is not obvious that this algorithm would converge, but our analysis shows that it does.
>
> (2) Our paper does not claim that Expectigrad has strictly better complexity than Yogi. Both algorithms have an asymptotic complexity of $O(1/b)$, technically speaking, due to their constant terms. However, it is true that our bound is lower _numerically_ than that of Yogi for large $T$-values since the constant $\epsilon$ is very small. We can see this by examining the relative magnitudes of the constant terms: for Expectigrad, this term is proportional to $1/\epsilon$ whereas Yogi's term is proportional to $1/\epsilon^2$. This result and our smoothness assumptions imply that Expectigrad will be closer, on average, to a stationary point than Yogi after training for a long time.
>
> The confusion here appears to be the $O(1/\sqrt{T})$ term in Expectigrad's bound. We obtained this rate because of the coefficient $\sqrt{1 - \beta_2} = 1/\sqrt{t}$ produced by Expectigrad's denominator. In the Yogi proof, this coefficient is constant ($\beta_2$ is a fixed value) and so the constant term was merged with the $O(1/b)$ term to allow the increasing minibatch size to anneal it over time. This does not mean that Expectigrad's complexity is worse than Yogi's, as we could do the same and achieve the same apparent rate of $O(1/T + 1/b)$.
>
> (3) Thank you for pointing out the inaccuracy in our footnote. We can update this in the final paper.

---

### Decision · Program_Chairs · 2021-01-07
**Final Decision**

**Decision:**

Reject

**Comment:**

The paper proposes a new adaptive optimization algorithm which is claimed to have better convergence properties and lower susceptibility to gradient variance. Reviewers found the idea of normalizing on the fly to be interesting, but raised some important concerns. Although similar to AdaGrad, Expectigrad has a very important differentiation due to division by $n_t$. Assuming $\beta=0$ in my opinion is also ok and many papers assume this for analysis. Even after accounting for these two facts, during discussions the reviewers considered the work to be incremental and a more thorough evaluation is needed to determine the benefits of algorithm. Specifically, please compare to important and relevant baselines (like AdamNC and Yogi), because sometimes it felt like baselines were picked and dropped randomly. The empirical improvement provided by Expectigrad compared to SOTA is not clear (both on synthetic problems from Reddi et al and real problems). Thus, unfortunately, I cannot recommend an acceptance of the paper in the current form. However, I would strongly encourage authors to resubmit after improving according to reviewer suggestions.

Some other minor points that came up during discussion are:
1. choice of hyperparameters was not clear to reviewers, e.g. different optimizer may behave very differently for same set of hyperparameters, so it would not be fair to compare them as is.
2. gradients would never be exactly zero in deep networks, so is current definition of $n_t$ good enough?